# Resection Ratios and Tumor Eccentricity in Breast-Conserving Surgery Specimens for Surgical Accuracy Assessment

**DOI:** 10.3390/cancers16101813

**Published:** 2024-05-09

**Authors:** Dinusha Veluponnar, Behdad Dashtbozorg, Marcos Da Silva Guimaraes, Marie-Jeanne T. F. D. Vrancken Peeters, Lisanne L. de Boer, Theo J. M. Ruers

**Affiliations:** 1Department of Surgery, Netherlands Cancer Institute, Plesmanlaan 121, 1066 CX Amsterdam, The Netherlands; 2Department of Nanobiophysics, Faculty of Science and Technology, University of Twente, Drienerlolaan 5, 7522 NB Enschede, The Netherlands; 3Department of Pathology, Netherlands Cancer Institute, Plesmanlaan 121, 1066 CX Amsterdam, The Netherlands; 4Department of Surgery, Amsterdam University Medical Center, Meibergdreef 9, 1105 AZ Amsterdam, The Netherlands

**Keywords:** excision volume, resection ratio, breast-conserving surgery, breast cancer, cosmetic outcome

## Abstract

**Simple Summary:**

This study aims to define and calculate several specimen parameters that would allow to determine the surgical accuracy of breast-conserving surgeries (BCS) in a representative population of patients. These specimen parameters included the ratio of specimen volume to tumor volume with different optimum margin widths (edges containing healthy tissue) and the tumor eccentricity, which is a measure for how centrally the tumor is located in the excised specimen. When using a surgical margin width of 0 mm, 1 mm, 2 mm, and 10 mm, on average 19.16 (IQR 44.36), 9.94 (IQR 18.09), 6.06 (IQR 9.69) and 1.35 (IQR 1.78) times the ideal resection volume got excised, respectively. The median tumor eccentricity was 11.29 mm (SD = 3.99) and the median relative tumor eccentricity was 0.66 (SD = 2.22). These parameters could be used to compare surgical accuracy when evaluating new technologies for intraoperative BCS guidance in the future.

**Abstract:**

This study aims to evaluate several defined specimen parameters that would allow to determine the surgical accuracy of breast-conserving surgeries (BCS) in a representative population of patients. These specimen parameters could be used to compare surgical accuracy when using novel technologies for intra-operative BCS guidance in the future. Different specimen parameters were determined among 100 BCS patients, including the ratio of specimen volume to tumor volume (resection ratio) with different optimal margin widths (0 mm, 1 mm, 2 mm, and 10 mm). Furthermore, the tumor eccentricity [maximum tumor-margin distance − minimum tumor-margin distance] and the relative tumor eccentricity [tumor eccentricity ÷ pathological tumor diameter] were determined. Different patient subgroups were compared using Wilcoxon rank sum tests. When using a surgical margin width of 0 mm, 1 mm, 2 mm, and 10 mm, on average, 19.16 (IQR 44.36), 9.94 (IQR 18.09), 6.06 (IQR 9.69) and 1.35 (IQR 1.78) times the ideal resection volume was excised, respectively. The median tumor eccentricity among the entire patient population was 11.29 mm (SD = 3.99) and the median relative tumor eccentricity was 0.66 (SD = 2.22). Resection ratios based on different optimal margin widths (0 mm, 1 mm, 2 mm, and 10 mm) and the (relative) tumor eccentricity could be valuable outcome measures to evaluate the surgical accuracy of novel technologies for intra-operative BCS guidance.

## 1. Introduction

Breast-conserving therapy (BCT), which involves breast-conserving surgery (BCS) combined with adjuvant radiotherapy, is the treatment of choice for early-stage breast cancer [1,2]. In case of a non-palpable breast tumor, a breast localization device is placed within or adjacent to the tumor for the purpose of intra-operative guidance [3]. In current surgical practice, various forms of localization techniques are being used, including wire-guided localization (WGL) [4,5,6], radioactive localization (RSL or ROLL) [7,8,9,10,11,12,13,14], and intra-operative ultrasound [15,16,17]. If the tumor is completely removed, BCS leads to clinical outcomes equivalent to those of a mastectomy while preserving the breast, thereby leading to a better cosmetic outcome. Nevertheless, various studies report an unsatisfactory cosmetic outcome after BCS in up to 40% of patients [18,19,20]. An unsatisfactory cosmetic outcome could influence psychosocial functioning and could lead to a decreased quality of life [21,22,23]. These are significant health concerns, and therefore, factors that determine cosmetic outcome should be of importance when determining the surgical accuracy of breast-conserving surgeries. Overall, specimen volume is a statistically significant determinant of cosmetic outcome [24,25,26,27]. It is important to mention here that modern oncoplasty techniques allow to perform BCS even in patients with low breast volumes due to the possibility of immediate tissue volume restoration using flap reconstruction. However, specimen volume in relation to tumor volume remains a valuable outcome measure of the surgical accuracy of BCS.

The value of this outcome measure becomes even more apparent in light of the increasing number of technologies that are currently being investigated for the purpose of breast tumor localization. These techniques include magnetic and paramagnetic localization (Magseed [28,29,30,31], Sirius Pintuition [32,33], MOLLI [34], TAKUMI [35]), radiofrequency reflector-based localization (SAVI SCOUT [36,37,38]), radiofrequency identification tags (LOCalizer [39,40,41]), EnVisio [42]), and stereotactic marking using a carbon suspension [43,44,45]. Due to the lack of solid evidence with regard to clinical effectiveness, none of these investigated techniques have widespread adoption [46].

Overall, there is a lack of studies in which standardized, objective outcome measures are determined to evaluate the surgical accuracy of the mentioned novel techniques. Among all studies on new technologies for intra-operative BCS guidance, only the study of Zacharioudakis et al. focused on the tumor-to-specimen volume ratio in Magseed-guided surgeries (N = 100) compared to WGL (N = 100) but found no significant differences between both groups. As mentioned earlier, specimen volume in relation to tumor volume could be a valuable outcome measure for surgical accuracy.

A few studies have been published that focus on specimen volume in relation to tumor volume among BCS patients [47,48,49,50,51]. These studies used the calculated resection ratio (CRR) as an outcome parameter. The CRR is defined as the ratio of the specimen volume to the optimal resection volume, which is the tumor volume with an added volume of an arbitrarily chosen margin of healthy breast tissue [47,48,49,50,51,52]. An ideal BCS would yield a CRR of 1.0, which means that the resected specimen volume is equal to the optimal resection volume. All above-mentioned studies have considered a healthy margin of 10 mm, and the reported median CRR in all studies was higher than 1.0. The included patients that varied from only patients with palpable breast cancer, only patients with non-palpable breast cancer, to patients with either type of breast cancer. Furthermore, the technique for guidance during BCS differed in each study, and included wire localization, ultrasound guidance and radio-guided occult lesion localization. Haloua et al. conducted the largest study, in which they found a median CRR of 2.32 (SD 3.23) based on 9276 pathology excerpts in a nationwide registry in the Netherlands [48]. These pathology excerpts originated from patients who underwent BCS with either of the mentioned guidance methods, in the years 2012 and 2013 in the Netherlands [48]. The second largest study by Krekel et al. reported a median CRR of 2.5 (range 0.01–42.93) among 726 patients, who underwent BCS from 2006 to 2009 in four institutions in the Netherlands [49]. The smallest median CRR (1.0) among all studies was reported by Krekel et al. in a study on 30 patients, who underwent BCS with intra-operative guidance using ultrasonography. It is noteworthy to mention that the last study had the smallest study population of all studies, and only included patients with palpable breast cancer. Overall, the reported median CRR among all studies ranged between 1.0 and 4.8 [47,48,49,50,51,52]. Stated differently, on average, up to 4.8 times the optimal tissue volume was resected when considering a healthy margin of 10 mm as the optimum. Although most of the mentioned studies have used clear study methods to evaluate the CRR among large BCS patient populations [48,49,51,52], there are some limitations in the methods, which require further investigation.

One limitation is that all studies used an optimal resection volume based on an added margin of 10 mm since such a margin width is considered to be surgically feasible [47,48,49,50,51,52]. However, various studies have found that a resection margin of 1 mm [53,54,55] or 2 mm [56,57] would lead to positive clinical outcomes, and wider margins are not necessary. None of the studies considered an optimal resection volume based on a margin of 1 mm or 2 mm, which would yield optimal clinical and cosmetic outcomes, and therefore would be a more relevant margin width for determining surgical accuracy.

Furthermore, all studies excluded patients with ductal carcinoma in situ (DCIS), patients with invasive carcinoma of no special type (IC NST) combined with DCIS and patients with multifocal disease. However, it is important to investigate the surgical accuracy in these patient groups since they form a substantial part of the total group of BCS patients, and their lesions could be fairly more complex to excise. Additionally, the studies excluded patients with neoadjuvant systemic therapy (NST). The surgeries of these patients could also be complex since it could be challenging for surgeons to predict the tumor response and estimate the amount of tissue that needs to be resected. The response to treatment in patients with NST is usually monitored before surgery using magnetic resonance imaging (MRI), mammography, or ultrasound (US) imaging [58]. However, the assessment of tumor response based on any of these imaging modalities is not entirely concordant with the tumor response assessed at pathological evaluation. Lastly, it is important to note that the aforementioned groups were most probably excluded from all studies because calculating resection ratios for these patients would have been challenging, given the inability to assume a spheroid shape for these lesions.

Another limitation of all studies is that the researchers either derived the specimen volume from the specimen weight or assumed that the specimen volume was a sphere or ellipsoid with a radius equal to half the maximum specimen diameter mentioned in the pathology report. All researchers also assumed that the tumor itself was a sphere or ellipsoid to allow simple formula calculations. However, breast tumors [59], as well as BCS specimens [60], could have a very irregular shape. Therefore, both the specimen volume and tumor volume may not be accurately calculated based on these formulas. More accurate methods are required for estimating both parameters. Lastly, none of the studies investigated any parameter indicating how centrally the tumor is located in each specimen, which would be useful for determining overall surgical accuracy.

In this study, we aim to define and calculate six specimen parameters that would allow the assessment of surgical accuracy in a patient population more representative of all BCS patients, while overcoming the aforementioned limitations. It is hypothesized that these parameters could constitute quantitative and accurate methods for surgical accuracy assessment, thereby enabling a comparison of novel technologies designed for intra-operative BCS guidance.

## 2. Materials and Methods

### 2.1. Patient Selection

In this retrospective study, 100 patients from the N19BOR study cohort were selected. The N19BOR study is a prospective, non-randomized, cohort study, conducted from 2019 to 2023 at the Netherlands Cancer Institute. For this study, patients who underwent BCS at the Netherlands Cancer Institute-Antoni van Leeuwenhoek Hospital (NKI-AvL) from 2020 to 2023 due to invasive carcinoma (IC) and/or ductal carcinoma in situ (DCIS) were included. The study protocol was approved by the Institutional Review Board (IRBm 20-077). According to the medical research involving human subjects act, no written consent was required. The inclusion was not consecutive, as patients with a radiologic complete response (rCR) after NST were excluded as well as patients with an oncoplastic breast reduction. The patients that were selected from the N19BOR cohort for this particular study were 100 patients with a similar distribution of pathological diagnoses and neoadjuvant therapy to all BCS patients who underwent breast-conserving surgery at the NKI in the year 2021.

### 2.2. Pathology Processing

Each BCS specimen was processed into hematoxylin and eosin-stained (H&E) sections according to a standard protocol at the pathology department. According to this protocol, after receiving a fresh BCS specimen, the resection margins are inked using different colors on different sides. This allows for orientation during the subsequent macroscopic and microscopic examination. The inking is also important for determining the actual specimen surface during microscopic evaluation, which is essential for accurate margin assessment. Then, the specimen is frozen, after which it is serially sliced at approximately 3 mm intervals from the side oriented towards the nipple to the peripheral side. The first and last tissue slices are always processed into cellular thin sections. In addition, the interlaying slices are processed into cellular thin sections, especially if they contain tumor tissue according to a macroscopic evaluation by the pathological assistant. Thereafter, the thin sections are stained for microscopic evaluation. An overview of the pathology processing method is shown in Figure 1. During the microscopic examination, the margins are evaluated. According to Dutch guidelines, a positive margin status is defined as IC cells reaching the inked margin over a trajectory > 4 mm or DCIS cells reaching the inked margin over any trajectory [61]. In addition, focally positive margins are defined as IC cells reaching the inked margin over a trajectory ≤ 4 mm) [61]. Otherwise, the margins are considered negative [61]. On the other hand, according to the guidelines of the Society of Surgical Oncology (SSO) and the American Society for Radiation Oncology (ASTRO), a margin is considered tumor-positive when ink touches the invasive carcinoma or when there is DCIS present within 2 mm from the resection margin [62].

### 2.3. Calculation of Tissue Areas and Tumor-Margin Distances

All calculations were based on the digitized, pathological H&E sections of specimens. An experienced pathologist annotated all tumor areas of IC and DCIS in the H&E images. For the calculations, it was assumed that all tissue slices of which the H&E sections originated had approximately the same thickness and same tissue composition over the entire slice thickness. Furthermore, it was assumed that any interlaying section of a specimen that was not processed into an H&E section based on macroscopic evaluation only contained healthy breast tissue. All data analyses were performed using MATLAB (2022a, MathWorks Inc., Natick, MA, USA).

The resection ratio (RR) of a specimen was defined as the specimen volume divided by the tumor volume. First, we calculated the total resected tissue surface area ($A_{tissue}$) and the total tumor surface area ($A_{tumor}$) for each specimen. Since all slices were assumed to have an equal thickness, calculating specimen volume or tumor volume would mean multiplying both areas by the same thickness factor. This step was redundant since we calculated the ratio of specimen volume to tumor volume afterwards. Therefore, the RR of a specimen was calculated by dividing the total $A_{tissue}$ by the total $A_{tumor}$ over all slices. In order to calculate $A_{tissue}$, each H&E section was segmented, and the area (number of pixels) was computed using the “Area of Object” function (*bwarea*) in MATLAB. The $A_{tissue}$ of interlaying slices that were not processed into H&E sections were calculated by interpolating the computed $A_{tissue}$ of the adjacent tissue sections. The $A_{tumor}$ was calculated by segmenting the tumor areas annotated by the pathologist in each H&E image, with subsequent similar use of the *bwarea* function. The total $A_{tissue}$ and $A_{tumor}$ for the entire specimen were obtained by the summation of individual values calculated over all slices.

Furthermore, the maximum tumor–margin distance ($TMD_{max}$) and minimum tumor–margin distance ($TMD_{min}$) in mm were measured. If an H&E section contained multiple areas of DCIS and/or invasive carcinoma, the area of the convex envelope enclosing all tumor areas was calculated using the *bwconvhull* function in MATLAB. Figure 2 illustrates the method for calculating the $A_{tumor}$ and $TMD_{max}$ and $TMD_{min}$ in an H&E slice with one tumor area (a–c) and an H&E slice with multiple tumor areas (d–f).

### 2.4. Resection Ratio Parameters

The four resection ratio parameters that were calculated were (1) the median resection ratio (RR), (2) the median close RR ($RR_{Close}$), (3) the median wide RR ($RR_{Wide}$), and (4) the median planned RR ($RR_{Planned}$). The $RR_{Close}$ was defined as the $A_{tissue}$ divided by the $A_{tumor}$ with an added 1.0 mm margin of healthy tissue. In order to calculate the $A_{tumor}$ with the added margin, the entire tumor boundary in each H&E image was extended 1.0 mm in an outward direction using the Mask dilation function in MATLAB (*imdilate*). The $RR_{Wide}$ was defined as the $A_{tissue}$ divided by the $A_{tumor}$ with an added 2.0 mm margin of healthy tissue, and the $RR_{Planned}$ was defined as the $A_{tissue}$ divided by the $A_{tumor}$ with an added 10 mm margin of healthy tissue. The $RR_{Close}$, $RR_{Wide}$, and $RR_{Planned}$ are illustrated in Figure 3.

### 2.5. Tumor Eccentricity and Relative Tumor Eccentricity

The tumor eccentricity in each section was defined as $TMD_{max}-TMD_{min}$ in mm. This was determined for each H&E section of a BCS specimen and averaged. In an ideal excision, the tumor is centrally located in the specimen, and the tumor eccentricity would be close to zero mm. A higher tumor eccentricity means that the tumor is located further from the center of the specimen, as the maximum distance from the margin to the tumor is higher compared to the minimum distance to the tumor. However, if two specimens with different tumor diameters have an equal tumor eccentricity, the tumor with the smaller diameter will be more eccentrically located in the specimen compared to the larger tumor. Therefore, in order to compare tumor eccentricity between specimens with different lesion diameters, the relative tumor eccentricity was determined. This is defined as the tumor eccentricity divided by the pathological tumor diameter.

### 2.6. Statistical Analyses

All resection ratio parameters were compared among patient subgroups with different pathological diagnoses, with or without NST, and with different margin statuses. Wilcoxon rank sum tests were performed to test for significant differences between the median of the respective groups.

## 3. Results

### 3.1. Patient and Tumor Characteristics

The median patient age was 56 years (SD = 12.9) (Table 1). In total, 100 specimens were analyzed: 43 patients underwent primary surgery, 47 patients had received neoadjuvant chemotherapy (NACT) before surgery, and 10 patients had received neoadjuvant hormonal therapy (NAHT) before surgery (Table 1). The median lesion diameter was 15 mm (range 3 mm–70 mm) (Table 1). Histopathological examination of the specimens showed pure DCIS in 11 patients, IC NST in 41 patients, IC NST combined with DCIS in 37 patients, and ILC in 11 patients (Table 1). Regarding the margin status, according to Dutch guidelines, 15 patients had a positive margin status (Table 1), 10 of whom had positive margins due to IC cells and 5 patients had positive margins due to DCIS cells. In total, 7 patients had focally positive margins, and 78 patients had negative margins (Table 1). When considering the SSO-ASTRO definitions, 45 patients had positive margins, 17 of whom had positive margins due to invasive carcinoma cells in the resection margins, while 28 patients had DCIS present within 2 mm from the margin. It is important to mention again that all patients with a complete radiologic response were excluded in this patient population.

### 3.2. Resection Ratio Parameters

The calculated RR, $RR_{Close}$, $RR_{Wide}$, and $RR_{Planned}$ are shown in Figure 4. In this plot, the median, minimum, maximum, first quartile, and third quartile of the mentioned resection ratio parameters are displayed. It should be mentioned that the scale of this plot is logarithmic. The median RR among all patients is 19.16 (IQR 44.36). When the optimal margin width is defined as 1 mm ($RR_{Close}$), and 2 mm ($RR_{Wide}$), the median ratio between the optimal resection volume and tumor volume is 9.94 (IQR 18.09), and 6.06 (IQR 9.69), respectively. Thus, almost ten times more healthy tissue is resected when considering a margin of 1 mm, and approximately six times the amount of healthy tissue is resected considering a safe margin of 2 mm. Furthermore, it is noteworthy to mention that the use of a surgically feasible margin width of 10 mm causes a median $RR_{Planned}$ of 1.35 (IQR 1.78), meaning that on average, 35% of the resected volume consists of excessively resected healthy tissue in this patient population. The $RR_{Planned}$ ranges from 0.45 to 24.23. Overall, the ranges of all specimen parameters are quite large with multiple outliers.

#### 3.2.1. Different Pathological Diagnoses

Further analysis showed that all median resection ratio parameters were highest for the patient group with ILC (N = 11), while all median resection ratio parameters were lowest for the patient group with DCIS (N = 11) (Table 2). Furthermore, the patient group with IC NST combined with DCIS (N = 37) had higher median resection ratio parameters with substantially larger IQRs, compared to the patient group with only IC NST (N = 41) (Table 2). This trend was seen independent of the optimal margin width that was added. Even when the maximum margin width of 10 mm was used, the median excessive volume resection in patients with IC NST combined with DCIS was 56%, compared to 29% among patients with only IC NST (Table 2). However, the sizes of the individual subgroups were small, and none of the differences between the median resection ratio parameters of all subgroups were significant at the 5% significance level according to Wilcoxon rank sum tests.

#### 3.2.2. NST with High Response Compared to NST with Low Response or No NST

The patient group who received NST (N = 57) was divided into a group with high response to therapy ($NST_{HR}$) (N = 33), meaning a residual tumor percentage of < 50%, and a group with a low response to therapy ($NST_{LR}$) (N = 24), meaning a residual tumor percentage of ≥ 50%. In the high response group, there were 13 patients with IC NST, 18 patients with a combination of IC NST and DCIS, and 2 patients with ILC. In the low response group, there were 13 patients with IC NST, 9 patients with IC NST combined with DCIS, and 2 patients with ILC. The $NST_{HR}$ group had higher median resection ratio parameters with substantially larger IQRs; compared to the $NST_{LR}$ group and the patient group without NST (N = 43) (Table 3), Wilcoxon rank sum tests indicated a significant difference (*p*-value < 0.05) between all parameters of the $NST_{HR}$ group compared to the $NST_{LR}$ group (*p*-value < 0.05) as well as all parameters of the $NST_{HR}$ group compared to the group without NST (*p*-value < 0.05). All median resection parameters of the patient group without NST were higher than the $NST_{LR}$ group, although none of the differences were statistically significant (Table 3).

#### 3.2.3. Different Margin Status

There were 22 BCS specimens from which one or more H&E sections presented a $TMD_{min}$ equal to zero (positive or focally positive margins according to Dutch guidelines) (Table 4). These specimens had lower median resection ratio parameters compared to the specimens with negative margins (Table 4). The group of patients with positive margins consisted of two patients with DCIS, nine patients with IC NST, seven patients with IC NST combined with DCIS, and four patients with ILC. On the other hand, the patient group with negative margins consisted of 9 patients with DCIS, 32 patients with IC NST, 30 patients with IC NST combined with DCIS, and 7 patients with ILC. We performed Wilcoxon rank sum tests to compare the median resection ratios of specimens with a (focally) positive margin to those of specimens with negative margins, using the different definitions of optimal margin widths. Except for the median $RR_{Planned}$, we did not find any significant differences (*p*-value < 0.05) between both groups. It is noteworthy to mention that even among patients with positive margins, there was an excessive resection of healthy breast tissue, regardless of the used optimal margin width. Even the median $RR_{Planned}$ (surgically feasible margin width of 10 mm) of the specimens with positive margins was 1.16, indicating an average, excessive volume resection of 16% in our study population (Table 4).

### 3.3. Tumor Eccentricity

Figure 5a shows that the tumor eccentricity among the entire patient group ranged between 4.12 mm and 28.00 mm. The median eccentricity of all patients was 11.29 mm (SD = 3.99), meaning that on average, the $TMD_{max}-TMD_{min}$ was 11.29 mm.

The median relative tumor eccentricity was 0.66 (SD = 2.22). When investigating the relative tumor eccentricity, it was found that a tumor diameter below 10 mm led to a substantially higher relative eccentricity compared to a tumor diameter greater than 10 mm as can be seen in Figure 5b. The median tumor diameter of the patients with a $TMD_{min}$ of zero was 15.0 mm, and 15.5 mm for the patients with negative margins. The median tumor eccentricity among the patients with positive margins was slightly higher (13.4 mm, SD = 3.97), compared to the patients with negative margins (10.3 mm, SD = 3.98). Furthermore, the median relative tumor eccentricity among the patients with positive margins was also slightly higher (0.68, SD = 0.79), compared to the patients with negative margins (0.64, SD = 2.47). Both differences were not statistically significant.

## 4. Discussion

The current focus on innovative technologies for intra-operative guidance in the field of breast cancer surgery is on improving surgical accuracy in order to obtain complete tumor resection as well as limited resection of healthy breast tissue [46]. Objective parameters to measure the outcome of such technologies with regard to surgical accuracy are limited. Only a few studies report on the specimen volume in relation to the tumor volume, which is a key determinant of the cosmetic outcome and a valuable measure to evaluate surgical accuracy. In general, these studies have some inherent limitations to their study method, which causes the reported results to be less accurate and less representative of the entire BCS patient population.

In this study, we aimed to define and evaluate several specimen parameters that would allow to determine the surgical accuracy of breast-conserving surgeries in a study population more representative of the entire BCS patient population.

We tested the specimen parameters in a small patient population for our study. The study results indicate that when using a margin width of 1 mm or 2 mm, on average, 9.94 (IQR 18.09), respectively, 6.06 (IQR 9.69) times the ideal resection volume was excised in our study population. When using a margin width of 10 mm, on average 1.35 times the surgically planned resection volume was removed, which comes down to 35% of excessive tissue volume. All resection ratio parameters have high interquartile ranges, indicating the large dispersion of the values compared to the median.

When looking at the correlation between margin status and resection ratio parameters, it became evident that all parameters of the specimens with negative margins were higher than the parameters of specimens with positive margins in our study population. Nevertheless, all RR parameters among patients with positive margins were higher than 1.0. When using an optimal margin width of 10 mm, there was a median excessive volume resection of 16% among the patients with positive margins. The median relative tumor eccentricity among the patients with positive margins was slightly higher (0.68), compared to the patients with negative margins (0.64), although not significantly different. Overall, there was a small difference in resection ratio parameters and (relative) tumor eccentricity between patients with a positive margin status compared to patients with a negative margin status.

Another finding was that on average the excessive tissue removal was the highest for the patient group with ILC or ILC combined with LCIS. A possible explanation for this occurrence is that ILC is associated with a higher likelihood of multifocal disease and a higher positive margin rate [63,64], which might cause breast surgeons to excise a larger margin of healthy tissue with the idea of preventing positive margins. On the other hand, all resection ratio parameters were lowest for the patient group with only DCIS. This could be explained by three different causes. First, DCIS lesions often occur as multiple islets with various sizes and distances from each other. Since the $A_{tumor}$ in these H&E sections is calculated as a convex envelope enclosing all tumor areas, the interlaying healthy tissue is also part of the ‘$A_{tumor}$’ since these cannot be resected separately. Therefore, the resection ratio parameters become smaller. Second, since multiple biopsies are performed to diagnose DCIS, part of the lesions are already removed through the biopsy needles before surgery, which could cause surgeons to resect a smaller volume. In our data set, the median diameter of the DCIS lesions was 13 mm, while the median diameter of the IC lesions was 19 mm. Lastly, the number of DCIS patients in our study population is limited to 10 patients. Another important study result is that the patient group with IC NST combined with DCIS (N = 37) had larger resection ratio parameters compared to the patients with only IC NST. This could be explained by the fact that in 15 of the 37 patient cases, the breast surgeon was aware of the combined lesion before the surgery due to the biopsy results beforehand. This could have caused a larger planned excision volume than the usually planned surgical margin of 10 mm. As earlier mentioned, there were no statistically significant differences between the median resection ratio parameters of the subgroups with different pathological diagnoses.

Furthermore, we found that the patient group with NST with a high response had significantly higher resection ratio parameters compared to the group with a low response to NST (*p*-value < 0.05), respectively, the group without NST (*p*-value < 0.05). One possible explanation for this discrepancy might be that the residual tumor diameter visible on medical imaging after NST is frequently an overestimation of the actual residual pathological tumor diameter, which causes surgeons to overestimate the tumor size when planning the surgery. Another possible explanation is that in many cases, there is a certain time period between the last medical image of the lesion before surgery and the surgery itself, during which the NST causes continued shrinkage of the tumor.

Lastly, we found a median tumor eccentricity of 11.29 mm (SD = 3.99) in our study population. In addition, we found that a lower tumor diameter leads to a higher relative tumor eccentricity compared to a higher tumor diameter, especially when the tumor diameter is below 10 mm.

It should be mentioned that there are many factors that could influence these specimen parameters, including the pathological diagnosis (and a priori knowledge), type of NST, tumor response to NST, experience level of the surgeons, etc. Therefore, when these specimen parameters are used to determine surgical accuracy of novel localization techniques, a large patient population should be evaluated.

There are several limitations to this study that need to be acknowledged. First, some of the included BCS specimens had one or more sections that were not processed into H&E sections. It was assumed that none of these sections contained tumorous tissue, while in actuality, this might not have been the case. However, we expect that the unprocessed sections that did contain tumor tissue would have had very small volumes of tumor tissue since they were not detected during the macroscopic evaluation by experienced pathological assistants. An additional limitation was that each tissue slice was assumed to be approximately 3 mm thick and did not change in composition in depth, while in actuality, these assumptions may not be true. We consider the impact of these potential errors on the calculations to be minimal, and, therefore, the calculated specimen parameters are still a good representation of the actual values. Another limitation is that during pathological processing, there could be a difference in the level of tissue shrinkage of healthy, mostly fatty, breast tissue between the tumor edge and the margin, compared to the more dense tumor area beneath it. This could have an influence on all calculated specimen parameters.

## 5. Conclusions


In conclusion, the novel contributions of this study are as follows:


Firstly, we defined and evaluated six specimen parameters in a data set of 100 BCS patients, representative of the entire patient population, including patients with different patient, tumor, and neoadjuvant treatment characteristics.


Secondly, we developed accurate calculation methods for these specimen parameters, based on actual pathological dimensions of BCS specimens.

We found that when using a margin width of 1 mm or 2 mm, on average, 9.94 (IQR 18.1) and 6.06 (IQR 9.69) times the ideal and clinically feasible resection volume was excised in our patient population, respectively. When using a surgically feasible margin width of 10 mm, 35% of the median specimen volume consisted of excessive healthy tissue. Additionally, the median tumor eccentricity was 11.29 mm (SD = 3.99), and the median relative tumor eccentricity was 0.66 (SD = 2.22). The relative tumor eccentricity was substantially higher for patients with a tumor diameter below 10 mm.


Thirdly, we performed subgroup analyses to determine significant differences in these specimen parameters between patients with different pathological diagnoses, with or without NST, and with different margin statuses.

We found that among patients with positive margins, on average, 16% of the excised volume consisted of excessive healthy tissue. Moreover, the patient subgroup with a high response to NST had a significantly higher resected volume of healthy breast tissue compared to patients without NST or patients with a low response to NST (*p*-value < 0.05).


These study results show that the defined specimen parameters in this study could be used to compare the surgical accuracy between different technologies for intra-operative BCS guidance in a novel manner. These parameters could be quantitative and accurate methods for assessing surgical accuracy compared to other parameters such as positive margin rate, secondary surgery rate, or volume/weight of resected tissue. Therefore, they could also be used to evaluate the learning curve of surgeons when using novel surgical guidance techniques.

## Figures and Tables

**Figure 1 cancers-16-01813-f001:**
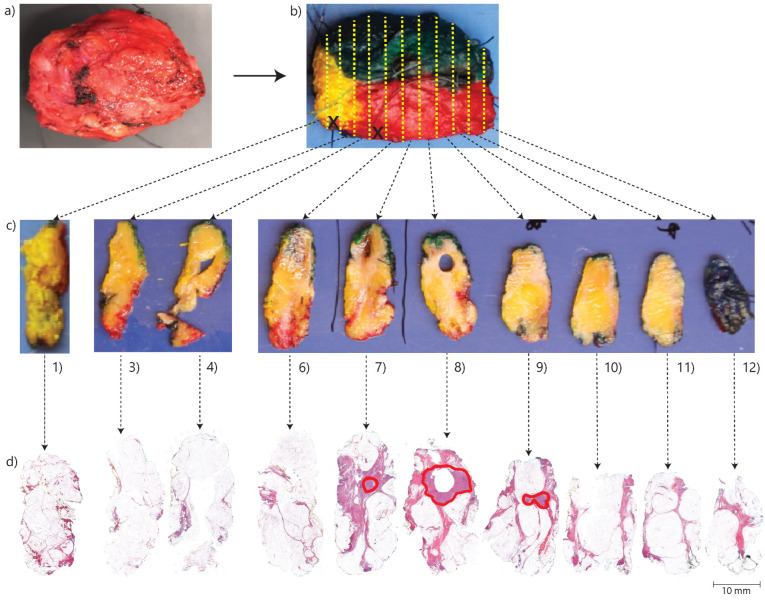
Overview of the pathology-processing method. In (**a**), a breast cancer specimen. In (**b**), the various resection margins are inked in different colors for orientation purposes according to the standard protocol. The yellow dashed lines indicate the intervals of slicing, which are approximately 3 mm. The specimen is cut along a plane that is perpendicular to the direction from the side facing the nipple (left side in photo) towards the peripheral side (right side in photo). The black crosses signify slices of which no H&E sections are made, in this case, the 2nd and 5th slices from the left side. In (**c**), a macro photo of the tissue slices, which are numbered 1 to 12 from left to right. In (**d**) H&E sections of the tissue slices. These sections are made by embedding the tissue slices in paraffin, after which they are cut into cellular thin sections of 0.003 mm by a microtome. The red annotations are locations with tumor tissue annotated by an experienced pathologist.

**Figure 2 cancers-16-01813-f002:**
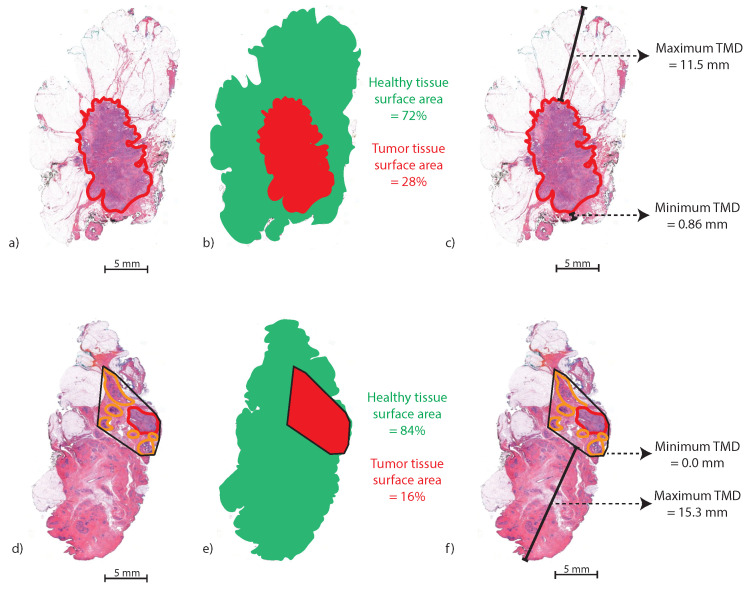
Method for calculating the $A_{tumor}$ in two H&E examples. (**a**) An H&E slice with one annotated IC area (red), (**b**) the segmented $A_{tumor}$ (red) and healthy tissue area (green), (**c**) $TMD_{max}$ and $TMD_{min}$ (black lines). (**d**) An H&E slice with one IC area (red) and multiple DCIS areas (orange). (**e**) The segmented $A_{tumor}$ (red), which is the convex envelope around the separate tumor areas. (**f**) The $TMD_{max}$ and $TMD_{min}$ of the convex envelope.

**Figure 3 cancers-16-01813-f003:**
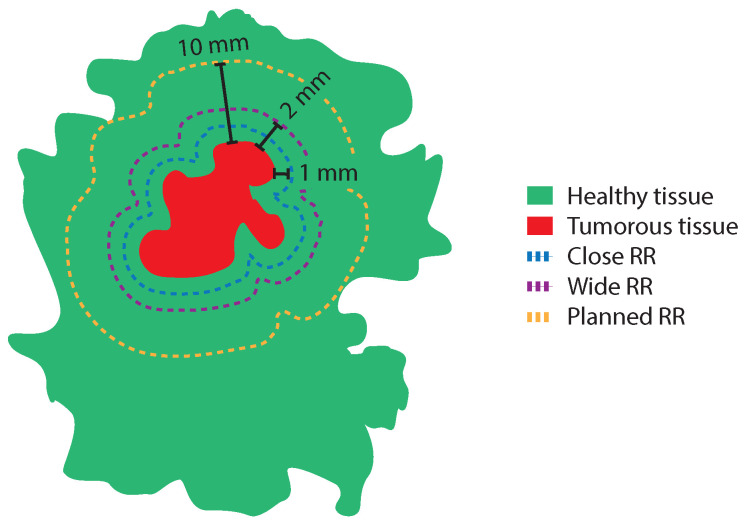
A schematic overview of a BCS specimen with tumor tissue (red) surrounded by healthy tissue (green), and three different margin widths for different optimal resection volumes. These include $RR_{Close}$ (blue dashed line), $RR_{Wide}$ (purple dashed line), and $RR_{Planned}$ (yellow dashed line).

**Figure 4 cancers-16-01813-f004:**
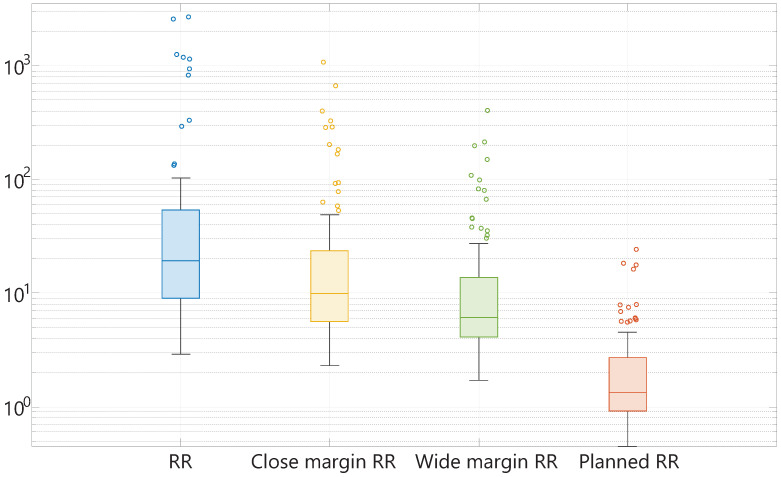
From left to right (with different colors), the box plots of the RR, $RR_{Close}$, $RR_{Wide}$, and $RR_{Planned}$, displaying the median, minimum, maximum, first quartile, and third quartile of each parameter on a logarithmic scale. The small circles represent the outliers.

**Figure 5 cancers-16-01813-f005:**
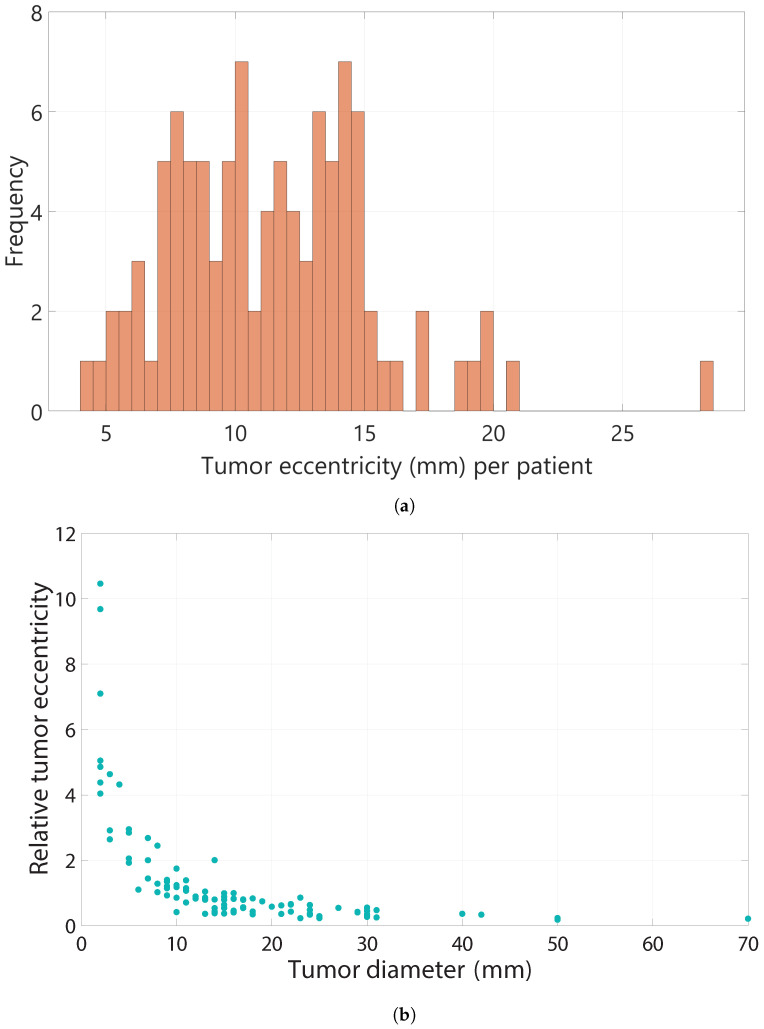
(**a**) Frequency of different values of tumor eccentricity among all patients. (**b**) Relative tumor eccentricity for different tumor diameters (mm).

**Table 1 cancers-16-01813-t001:** Patient and tumor characteristics.

	Characteristic N = 100
**Age** (years) (median, SD)	56 (12.9)
**Pathological lesion size at pathological evaluation** (mm) (median, min, max)	15 (3, 70)
**Specimen weight** (gram) (median, SD)	18 (17)
**Histological tumor type at pathological evaluation**	
DCIS	11
IC NST	41
IC NST + DCIS	37
ILC	11
**T-stage**	
pTis	11
pT1a	11
pT1b	22
pT1c	39
pT2	14
pT3	3
**Histological tumor grade at pathological evaluation**	
1	19
2	46
3	35
**Hormonal receptor and HER2 status**	
ER+/PR+ HER2−	49
ER+/PR+ HER2+	36
ER−/PR− HER2+	3
TN	12
**Neoadjuvant treatment**	
Chemotherapy	8
Chemotherapy and immunotherapy	39
Endocrine therapy	10
None	43
**Margin status ***	
Negative	78
Focally positive	7
Positive	15

IC NST = invasive carcinoma of no special type, DCIS = ductal carcinoma in situ, ILC = invasive lobular carcinoma; ER = estrogen receptor, PR = progesterone receptor, TN = triple negative, * = according to Dutch guidelines.

**Table 2 cancers-16-01813-t002:** Resection ratio parameters for subgroups with different pathological diagnoses.

	RR (IQR)	${\mathrm{RR}}_{\mathrm{Close}}$ (IQR)	${\mathrm{RR}}_{\mathrm{Wide}}$ (IQR)	${\mathrm{RR}}_{\mathrm{Planned}}$ (IQR)
**DCIS (N = 11)**	17.19 (13.05)	8.86 (3.67)	5.25 (2.07)	1.05 (0.47)
**IC NST (N = 41)**	19.07 (44.09)	10.71 (16.87)	6.66 (9.51)	1.29 (1.84)
**IC NST and DCIS (N = 37)**	20.40 (81.97)	11.23 (41.88)	7.27 (23.97)	1.56 (2.84)
**ILC (N = 11)**	24.81 (30.20)	14.20 (8.81)	9.09 (4.58)	1.69 (0.87)

**Table 3 cancers-16-01813-t003:** Resection parameters for the subgroups with NST and the subgroup without NST.

	RR (IQR)	${\mathrm{RR}}_{\mathrm{Close}}$ (IQR)	${\mathrm{RR}}_{\mathrm{Wide}}$ (IQR)	${\mathrm{RR}}_{\mathrm{Planned}}$ (IQR)
${\mathrm{NST}}_{\mathrm{HR}}$ **(N = 33)**	47.34 (284.46)	21.30 (82.82)	12.96 (39.36)	2.03 (4.43)
${\mathrm{NST}}_{\mathrm{LR}}$ **(N = 24)**	10.91 (11.60)	6.91 (4.64)	4.69 (1.82)	1.09 (0.62)
**No NST (N = 43)**	16.20 (16.12)	8.98 (9.89)	5.51 (5.36)	1.27 (1.17)

**Table 4 cancers-16-01813-t004:** Resection ratio parameters for patients with positive margins compared to patients with negative margins.

	RR (IQR)	${\mathrm{RR}}_{\mathrm{Close}}$ (IQR)	${\mathrm{RR}}_{\mathrm{Wide}}$ (IQR)	${\mathrm{RR}}_{\mathrm{Planned}}$ (IQR)
**Positive margins (N = 22) ^a^**	15.07 (18.36)	7.42 (9.53)	4.44 (4.58)	1.16 (0.84)
**Negative margins (N = 78)**	20.40 (45.93)	11.23 (24.36)	7.25 (14.12)	1.37 (1.70)

^a^ Positive margins are defined as focally positive margins, and more than focally positive margins.

## Data Availability

Data underlying the results presented in this paper are not publicly available at time but may be obtained from the authors upon reasonable request.

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
