# Peer review of "Resection Ratios and Tumor Eccentricity in Breast-Conserving Surgery Specimens for Surgical Accuracy Assessment"

_cancers, 2024, doi:10.3390/cancers16101813_

Round 1

Reviewer 1 Report

Comments and Suggestions for Authors

First of all, I would like to congratulate the authors for the article, which reflects the work and effort that must surely be behind it.

My general recommendation is that it should be published. However, I think it would be interesting to make some formal remarks that, in my opinion, would enrich the work and make it more easily understandable for researchers or clinicians interested in this topic.

Regarding the introduction:

It has to state clearly and concisely why this work is important. The first 3 paragraphs have managed to attract my interest as a clinician, they raise the current problem, and the authors make it clear that there is a lack of standardization of the procedure for the evaluation of surgical specimens. But, in the following paragraphs (4-7), on the other hand, there is a very detailed description of the results of different studies with a critical evaluation which, in my opinion, could fit more in the discussion section. In fact, it is in the discussion section where, the comparison of the technique and the results obtained by the researchers with those published by other authors, allows us to have a global vision and put the results into perspective. It is in the discussion section where the results obtained are interpreted, providing bibliography, making a critical judgement of it, comparing it with our results and explaining possible biases.

Finally, the last paragraph of the introduction should be used to summarize  the objective of the study, to state the working hypothesis. Instead, in this case, it begins to describe a "how did we do it?", which would be a "material and methods" section (“In this study, we have evaluated six specimen parameters that would allow surgical accuracy assessment in a patient population more representative of all BCS patients, while overcoming the aforementioned limitations. Four of the specimen parameters are calculated based on specimen volume to tumor volume ratios with different optimum margin widths, and two parameters represent how eccentrically the tumor is located in each BCS specimen”). 

Material and methods: Perfectly explained, congratulations. 

Results:

Perfectly explained and understandable.

However, consider that in section “3.2.3. Different margin status”, that describing in the text exactly all the data provided in table 4 is repetitive. When presenting the results, the tables should be an "extra" piece of information, a complement to what is described in the text, not an exact copy. In fact, table 4 has an "empty" look, giving the impression of a typographical error by having so many empty spaces in the columns. Consider completing it with the analysis of the variables (RR (IQR) RRClose (IQR) RRWide (IQR) RRPlanned (IQR)) performed for each subgroup (DCIS, IC NST, IC NST + DCIS, ILC) or, if only Positive margins (N = 22) and Negative margins (N = 78) are to be assessed, and since the data have been described in the text, consider deleting the table.

Discussion

The first paragraph would fit in the introduction section. It clearly shows the lack of objective evidence about the problem presented and poses a thread towards the presentation of a hypothesis.

The rest of the discussion is well organized, but I am missing the perspective of what other studies say about each of the results presented, whether they agree or disagree with those obtained by the authors and, if not, what is the critical reading of it and whether there is a possible explanation.

Results: Excellent

Reviewer 2 Report

Comments and Suggestions for Authors

I would like to cheer the authors for their interesting paper and the huge work that must stay behind such a big data collection and analysis.

Even if I may recommend it for publication, I would like to point out few possible improvement suggestions:

1. The title of the paper seems incomplete as it does not show the purpose of the research. Please complete it as "Resection ratios and tumor eccentricity in breast-conserving specimens for surgical accuracy assessment"

2. In page 2, Introduction section, the authors state: "These are significant health concerns, and therefore factors that determine cosmetic outcome should be of importance when determining the surgical accuracy of breast-conserving surgeries. Overall, specimen volume is a statistically significant determinant of cosmetic outcome [24–27]. Thus, specimen volume in relation to tumor volume is a valuable, outcome measure of the surgical accuracy of BCS." It is true that tumor resection volume is crucial for cosmetic outcome, especially in smaller size breasts. Nevertheless, modern oncoplasty techniques allow to push BCS even in small breasts, thanks to the possibility of tissue volume immediate restoration with local flaps as the ICAP ones. Please spesicy this briefly.

3. In the Conclusion section, the authors resume their results but a finer conclusion is lacking. Please, describe the conclusions of the study: what is novel from the study? What advantages may these parameters bring to the clinical practice?

Reviewer 3 Report

Comments and Suggestions for Authors

Here are my comments and suggestions on your manuscript: 

The main question addressed by the research is to evaluate specimen parameters that determine the surgical accuracy of breast-conserving surgeries (BCS) and their potential use in comparing surgical accuracy with novel technologies for intraoperative BCS guidance.

The topic is highly relevant in the field as it addresses the need for objective measures to assess surgical accuracy in breast-conserving surgeries, adding new results to currently existing literautre data. 

The study  contributes to the ongoing efforts to enhance surgical techniques and outcomes in breast cancer treatment.

Including a larger sample size or validating the findings with long-term clinical outcomes could further enhance the study's relevance.

The conclusions drawn by the authors are consistent with the evidence presented. 

Based on the findings of this study, I recommend the insertion in the introduction of one or two parragraphs related to imaging tumor assessment methods used in breast cancer patietns following neoadjuvant chemotherapy.
